# Seamless Mosaicking of UAV-Based Push-Broom Hyperspectral Images for Environment Monitoring

Lina Yi [1], Jing M. Chen [2,3,*], Guifeng Zhang [4,5], Xiao Xu [1], Xing Ming [6] and Wenji Guo [6]

1   School of Geoscience and Surveying Engineering, China University of Mining and Technology-Beijing, Ding 11 Xueyuan Road, Haidian District, Beijing 100083, China; linayi@cumtb.edu.cn (L.Y.); tutu3366@163.com (X.X.)

2   Department of Geography and Planning, University of Toronto, Toronto, ON M5S 3G3, Canada

3   School of Geographical Science, Fujian Normal University, No. 8 Shangsan Road, Cangshan District, Fuzhou 350007, China

4   Key Laboratory of Computational Optics Imaging Technology, Aerospace Information Research Institute, Chinese Academy of Sciences, No.9 Dengzhuang South Road, Haidian District, Beijing 100094, China; zhanggf@aircas.ac.cn

5   School of Opto-Electronics, University of Chinese Academy of Sciences, No.19(A) Yuquan Road, Shijingshan District, Beijing 100049, China

6   Nanjing Institute of Software Technology, Institute of software Chinese Academy of Sciences (ISCAS), Building 4, Artificial Intelligence Industrial Park, No. 266, Chuangyan Road, Kirin Science and Technology Park, Nanjing 210000, China; mingxing@nj.iscas.ac.cn (X.M.); wenji@nj.iscas.ac.cn (W.G.)

*   Correspondence: jing.chen@utoronto.ca

**Abstract:** This paper proposes a systematic image mosaicking methodology to produce hyperspectral image for environment monitoring using an emerging UAV-based push-broom hyperspectral imager. The suitability of alternative methods in each step is assessed by experiments of an urban scape, a river course and a forest study area. First, the hyperspectral image strips were acquired by sequentially stitching the UAV images acquired by push-broom scanning along each flight line. Next, direct geo-referencing was applied to each image strip to get initial geo-rectified result. Then, with ground control points, the curved surface spline function was used to transform the initial geo-rectified image strips to improve their geometrical accuracy. To further remove the displacement between pairs of image strips, an improved phase correlation (IPC) and a SIFT and RANSAC-based method (SR) were used in image registration. Finally, the weighted average and the best stitching image fusion method were used to remove the spectral differences between image strips and get the seamless mosaic. Experiment results showed that as the GCPs' number increases, the mosaicked image's geometrical accuracy increases. In image registration, there exists obvious edge information that can be accurately extracted from the urban scape and river course area; comparative results can be achieved by the IPC method with less time cost. However, for the ground objects with complex texture like forest, the edges extracted from the image is prone to be inaccurate and result in the failure of the IPC method, and only the SR method can get a good result. In image fusion, the best stitching fusion method can get seamless results for all three study areas. Whereas, the weighted average fusion method was only useful in eliminating the stitching line for the river course and forest areas but failed for the urban scape area due to the spectral heterogeneity of different ground objects. For different environment monitoring applications, the proposed methodology provides a practical solution to seamlessly mosaic UAV-based push-broom hyperspectral images with high geometrical accuracy and spectral fidelity.

**Keywords:** geometric rectification; image registration; image fusion; spectral fidelity

## 1. Introduction

Hyperspectral images have hundreds of narrow and nearly continuous spectral bands that represent the reflectance signals of the observed objects [1]. In early years the imaging

spectrometers were often used in multispectral mode due to the technical limitations of the instruments. Fine spatial and spectral resolution creates problems with the signal-to-noise ratio of the sensor used. The number of spectral bands had to be reduced in order to get smaller pixel size as reading the data from the sensor takes some time and using many spectral bands would have resulted in pixels elongated towards the flight direction because plane speed cannot be reduced below a certain critical limit [2]. Recent developments in the miniaturization of electro-optical sensors and unmanned aerial vehicles (UAVs) have established a new era of hyperspectral imaging [3]. Due to the high flexibility of UAV flight planning, hyperspectral images taken by UAVs can be widely used in environment monitoring applications [4–11], such as precision agriculture, species classification [11–13], surface parameter retrieval [14,15], etc.

The wider use of UAVs has also pushed also instrument developers. There are two main types of lightweight hyperspectral sensors for UAVs: push-broom scanners [16–18] and snapshot imagers [18–21]. Push-broom hyperspectral scanners are based on a linear array that captures scenes while operating in a push-broom mode and the acquired images have a high spectral sensibility [16]. Snapshot imaging sensors can obtain bundles of frames at the same time to form an image with two spatial dimensions and tens to hundreds of bands [18]. The push-broom sensor system outperforms snapshot image capturing approaches, as the latter systems require a compromise between spatial coverage, spatial resolution and spectral resolution [20]. Spectral resolution of push-broom hyperspectral scanners has improved in recent years. Most of the sensors have 5 nm or better spectral resolution and can be flown with 1 m or smaller pixel sizes. A UAV carrying a push-broom hyperspectral imager that is light weight and has a small size gradually becomes a payload for acquiring multi-temporal and high spatial resolution hyperspectral images.

In hyperspectral data acquisition, the UAV is usually operated at low altitudes, and the implementations always result in some geometric and radiometric distortions within the acquired data such as pixel displacement errors and spectral noises caused by smile effects. Thus there exists a lot of challenges in the post-processing of hyperspectral images for spectrally complex environments [3]. It is often necessary to obtain multiple UAV images to cover a study area with different overlaps between images. Mosaicking the UAV images into geometrical and radiometric high quality hyperspectral images is one of the most essential procedures to meet the application needs of remote sensing environment monitoring [22–24]. However, as the push-broom sensor system is an emerging technology that is of high cost, the research work on it is not popularized in science. Only a few research works have been published to mosaic the push-broom hyperspectral images mounted on a UAV [16–18].

The ZK-VNIR-FPG480 hyperspectral imaging system is an emerging push-broom sensor with independent property rights in China that offers a good combination of spatial and spectral resolution and has been proved effective in environment monitoring applications [25]. The image has a total of 270 bands, the spectral range is 400–1000 nm, the spectral resolution is 2.8 nm, and the spatial resolution is 0.9m at 1 km flight height. While no comprehensive tools are available for producing mosaics of UAV hyperspectral images acquired by this push-broom scanner, how to process and mosaic the images to get a seamless hyperspectral image is studied in this research.

Generally, the geometric and radiometric correction of the UAV hyperspectral images are two key steps in image mosaicking. In geometric correction, images obtained with different positions and attitudes should be mapped to an identical reference plane. In radiometric correction, the inconsistency in brightness between geometrically corrected images should be minimized, and then the radiometric corrected multiple images should be fused into a mosaicked image. The geometric quality of the mosaicked image is dependent on the geometric correction accuracy, while the radiometric quality of the image is impacted by the sensor calibration, radiometric correction and image fusion procedures.

### 1.1. Geometric Correction

In geometric correction, the geo-rectification and registration of UAV hyperspectral images are two key steps and have been active research fields recently [17,26–28].

In geo-rectification, each image acquired from an imaging system is tied to spatial locations relative to a desired reference frame with a geometric control. This control can be established using GCPs and/or the implementation of a Position and Orientation System (POS) [29]. Images can be rectified by global transformations derived from the relationships between GCPs and the corresponding image points [30] or by a direct Geo-reference scheme [31]. Push-broom scanners have an inherently unstable imaging geometry due to wind-related 3-dimensional motions of the imaging platform during data acquisition, and therefore their post-processing ideally requires accurate navigation data which are typically measured by a high-end direct georeferencing system based on global navigation satellite system (GNSS)/Inertial navigation system (INS) [32,33]. There are a lot of GNSS and INS systems developed by UAV companies all over the world, such as the Sky2 airborne GNSS receiver (Hi-target company in China, https://en.hi-target.com.cn/precision-uav-kits, accessed on 1 November 2021), mosaic-X5 GNSS module (septentrio company in European, https://www.septentrio.com/en/applications/uavandrobotics, accessed on 1 November 2021) etc. Choosing the right module that is optimal for UAV use is the key to successful products. The Sky2 differential GPS module adopts a high dynamic receiver board, which can achieve cm level differential positioning results even at a high speed of 100 km/h. Mosaic-X5 proved to be the ideal PPK-capable module to deliver the accuracy for survey-grade results. Given that the accurate POS information provides the possibility to get the direct Geo-referenced images, the location errors are unavoidable. Using GCPs can increase the geometric accuracy, however, the displacement between pairs of images always exists.

To get a seamless mosaicked image, the accurate tie points in the overlapped area between pairs of images should be extracted and the geometric differences of the tie points should be minimized during image registration. An algorithm was proposed to automatically combine multiple overlapping images for a scene by a 2-dimensional fast fourier transform to estimate the displacement between image pairs by accounting for image rotation and scale variation. In this research work, the selection of a suitable hyperspectral band is critical for best co-registration to deal with the data volume and processing complexity of hundreds of bands; due to the limitation of the computer's computational capabilities, the Pearson correlation coefficient (PCC) was computed for each band in the hypercube pair by the iterating step first, then the band with the highest PCC value was retained to the mosaicking process [34]. A modified Speeded-Up Robust Features matching method was used to find the tie points between the partially rectified hyperspectral images and the RGB orthophoto to mosaic the UAV images for agricultural monitoring [28]. The above two methods were validated on frame images for which the overlap rate is high. Whereas, in push-broom hyperspectral image acquisition, sufficient overlaps between adjacent images may not be ensured, especially when data collection efficiency is a priority for a large study area with the constraint of UAV battery duration. Moreover, in some natural environments where the target objects are mainly water and forest, there exist even less texture or distinct features that can be detected as the tie points due to the homogeneity of the spectral signatures. Although existing relative transformation models and resampling methods have been analyzed for narrow overlapped image mosaicking, the lack of tie points and weak stereo geometry makes it difficult to get accurately geo-registered images [23].

### 1.2. Radiometric Correction

Due to the influences of the light variation during a flight, caused by viewing geometry or wind, as well as instrument noise, spectral signatures consecutively acquired over the same area can vary significantly, and hence a stitching line would appear in mosaics of geometrically corrected images. Stitching lines are therefore an issue that should be addressed in order to obtain seamlessly mosaicked hyperspectral images. It is therefore

highly useful to develop methods to eliminate the spectral differences and improve the radiometric accuracy of mosaicked images [27].

One solution is to ensure the image data collected at different strips are radiometrically consistent and the obtained surface reflectance estimates are reliable. In recent research, the authors performed experiments to assess the accuracy of the Parrot Sequoia camera and the sunshine sensor to get an indication of whether the quality of the data collected is sufficient to create accurate reflectance maps. The developed workflow is evaluated against data from a handheld spectroradiometer, giving the highest correlation ($R^2$ = 0.99) for the normalized difference vegetation index (NDVI). For the individual wavelength bands, $R^2$ was 0.80–0.97 for the red-edge, near-infrared, and red bands. They stated that surface reflectance is not directly measured by the UAV-based cameras. When an image is captured, the image sensor in the camera records the radiant (light) received by each pixel as a digital number (DN). To convert the DN to surface reflectance, a radiometric correction must be performed by (1) applying sensor-related corrections to obtain the radiance received by the camera from the DN, and (2) converting the radiance received by the camera to surface reflectance [35].

Although the above rigorous radiometric correction method is preferred, in practice, another solution for eliminating the spectral distortions, namely relative radiometric correction, is used more frequently. For example, a radiometric block adjustment method was proposed for the UAV image datasets by processing of close-range 2D frame format images with a comprehensive weighting scheme for both solar elevations and reflectance panel observations. However, this method was sensitive to a priori values, in particular to the illumination level, so it is important to devise more accurate measurement methods for the incident irradiance [36]. In other research, the hyperspectral imagery was obtained using a snapshot hyperspectral sensor mounted on a UAV, the radiometric response linearity and radiometric response variations were quantitatively evaluated and a method to correct the radiometric response variation effect was proposed. Comparison with spectra measured with an ASD Field Spec Pro spectrometer (Analytical Spectral Devices, Boulder, CO, USA) was done, and the results show that the radiometric precision was 5% for bands between 500 and 945 nm, and the reflectance curve in the infrared spectral region did not decrease [19].

The focus of this study is on the processing of the push-broom hyperspectral UAV images to acquire a seamless image that has a high geometric accuracy and spectral fidelity. A practical image mosaicking chain of methods that are applicable to different types of land surfaces was proposed. Several methods for image rectification, image registration and image fusion were compared and assessed by using a new set of UAV data acquired in urban scape, river course and forest areas in experiments. Results show that the proposed image mosaicking methodology can not only produce a hyperspectral image with a high image geometrical accuracy but also ensure the spectral fidelity before and after image mosaicking. The following paper is organized as follows: Section 2 introduces the data acquisition and preprocessing principles of the UAV-mounted hyperspectral imagery used in this study and proposes the image mosaicking methodology with comprehensive processing steps including geometric rectification, image registration and fusion. Section 3 presents experimental results and discusses the best strategy for each study area. Section 4 draws conclusions and proposes future work.

## 2. Materials and Methods

### 2.1. Data Acquisition

The ZK-VNIR-FPG480 hyperspectral imaging system (ZKYD Data Technology Co., Ltd., Beijing, China) (Figure 1a) includes an six-rotor UAV platform (DJI M600 Pro), an image stabilization platform, a hyperspectral imager, a high-speed data acquisition controller, and a POS. The hyperspectral imager and the Sky2 airborne GNSS receiver were pointed vertically downward on the drone DJI M600 Pro (Figure 1b). The imager is at the center of rotation of the gimbal, which is used to keep the sensor horizontal when

the UAV is moving. Table 1 shows a summary of the UAV hyperspectral imaging system properties. The sensor covers the spectral range of 400–1000 nm with a spectral resolution of 2.8 nm and a spatial resolution of 0.9 m at 1 km flying height. The max framerate, which is the maximum lines of data that are acquired in one second during flight, is 100 frames per second.

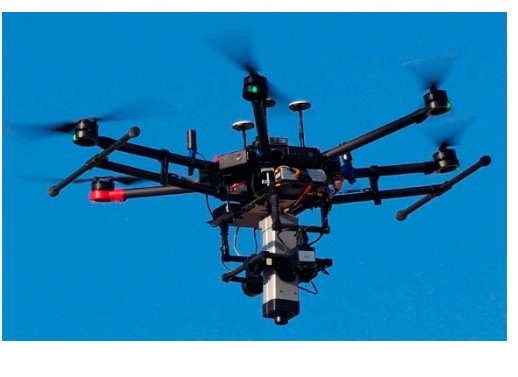
(**a**)

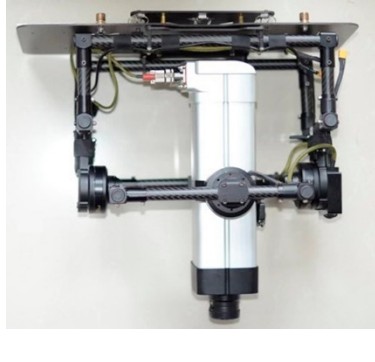
(**b**)

**Figure 1.** UAV hyperspectral system used in the study: (**a**) M600 Pro UAV platform with the hyperspectral imaging system onboard; (**b**) The ZK-VNIR-FPG480 hyperspectral imaging spectrometry system.

**Table 1.** Technical indicators of ZK-VNIR-FPG480 hyperspectral imager.

| Parameter | Indicators |
| --- | --- |
| Spectral range | 400 nm~1000 nm |
| Number of spectral channels | 270 |
| Spectral resolution | 2.8 nm |
| Line array width | 480 |
| Spatial resolution | 9 cm at 100 m |
| Field of View | 26° at 35 mm (lens focal length) |
| A/D conversion | 12 bits |
| Max framerate | 100 fps |

The UAV-based hyperspectral images are acquired at low altitudes in sunny weather. While the influence of the atmosphere on the images can be almost entirely ignored, a whiteboard with known reflectance data was put as the calibration target to facilitate the conversion of UAV image DN to surface reflectance [26]. To obtain as large an area of image data as possible during one flight, the flight lines were set parallel to acquire the hyperspectral image strips. In each flight line, linear push-broom detectors were used to scan the ground surface along the flight direction with 480 spatial pixels across the flight line, and the spectral signals were recorded and decomposed into 270 bands to construct the arrays of each hyperspectral image. At the same time of imaging, the Sky2 and GNSS satellite receiving antenna records the GNSS base's real-time kinematics (RTK) signals that are used to derive the UAV external orientation elements. The UAV turned around after reaching the terminal and flew along the next line until the entire study area was scanned.

In different areas, there are different limitations for UAV flights in China; the legislation for the UAV flights in China is published in 2015 by the Civil Aviation Administration of China [37]. As the flight height becomes higher, the photograph scale becomes smaller, the coverage of the area that can be captured in one scanning strip is larger and the number of strips of image that is needed to cover the study area become fewer. In the experiment, data collection of the three typical study areas are implemented. To reduce the time that is needed for data collection as well as consider the latter convenience of image mosaicking, different flight parameters and side overlap rates are set according to the characteristics of the study area before the flight implementation (Table 2).

**Table 2.** Flight parameters of different study areas.

| Study Area | Flight Height | Side Overlap Rate |
| --- | --- | --- |
| Urban scape | 200 m | 30% |
| River course | 120 m | 50% |
| Forest | 90 m | 20% |

As shown in Figure 2, the locations of the study areas are labelled using the red polygon. The urban scape area (960 m∗300 m) is a place near Erhai lake in Yunnan Province which is located in the southwest of China. When collecting the hyperspectral images, the UAV flight height is set at 200 m. From Figure 3a, it shows that the urban scape area is composed of buildings, roads, trees, farmland and a lake; it is easy to extract multiple high-precision tie points for image mosaicking, so the side overlap rate is set as 30%. The river course (1100 m∗68 m) is a part of Zhang wei xin river in DeZhou distinct of Shandong Province, which is located in the east of China, the flight height is set as 120m. The area is dominated by water (Figure 3b) where the spectral reflectance is homogeneous so it is difficult to find enough high-precision tie points. Moreover, the width of the river course is about 76% of the width of image strip; therefore, the side overlap rate was set at 50% to make sure there exist some distinct feature points that can be used in image registration. The forest area (570 m∗135 m) is an area that is located in Zhan jiang mangrove forest protection area in Guang dong Province, which is located in the south of China. As shown in Figure 3c, it is representative of mangrove, riparian plants and water. To get a high spatial resolution image for mangrove forest monitoring, the flight height is set as 90 m, so the coverage of each image strip is relatively small. In this situation, the side overlap rate was set at 20% to improve the data collection efficiency.

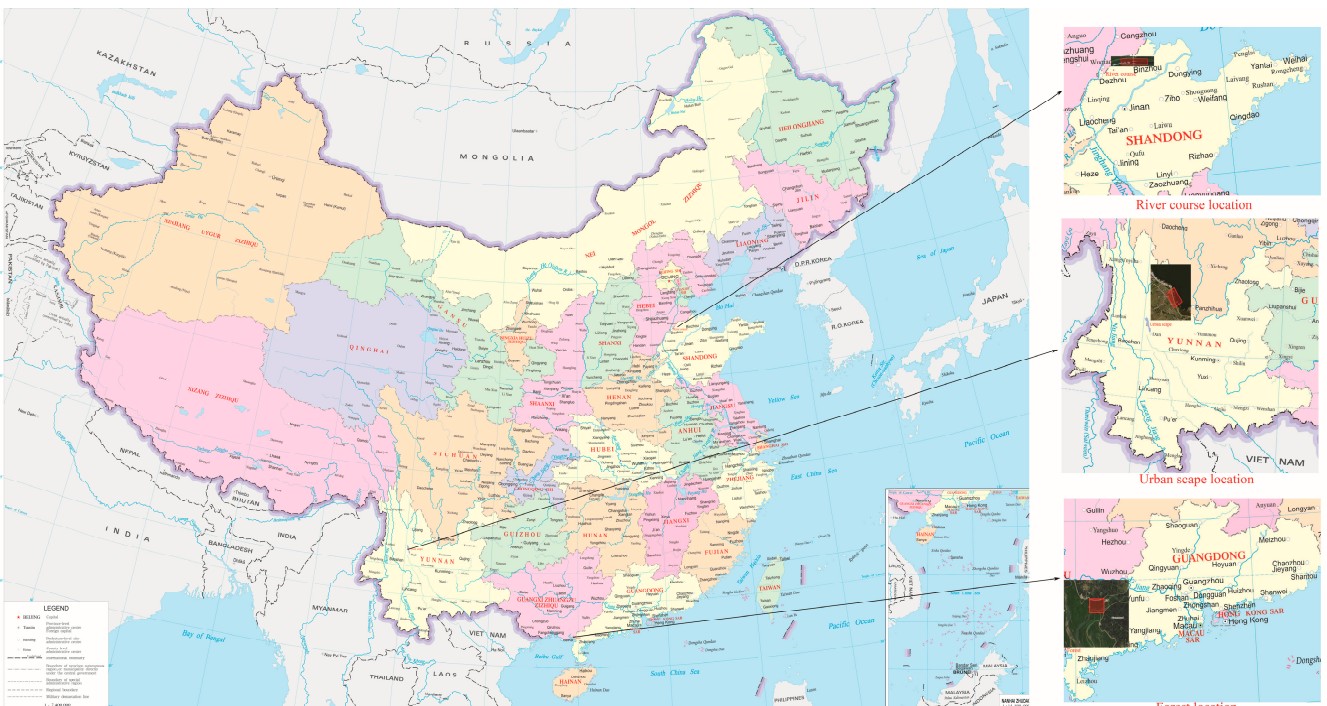

**Figure 2.** Location of the study areas in China.

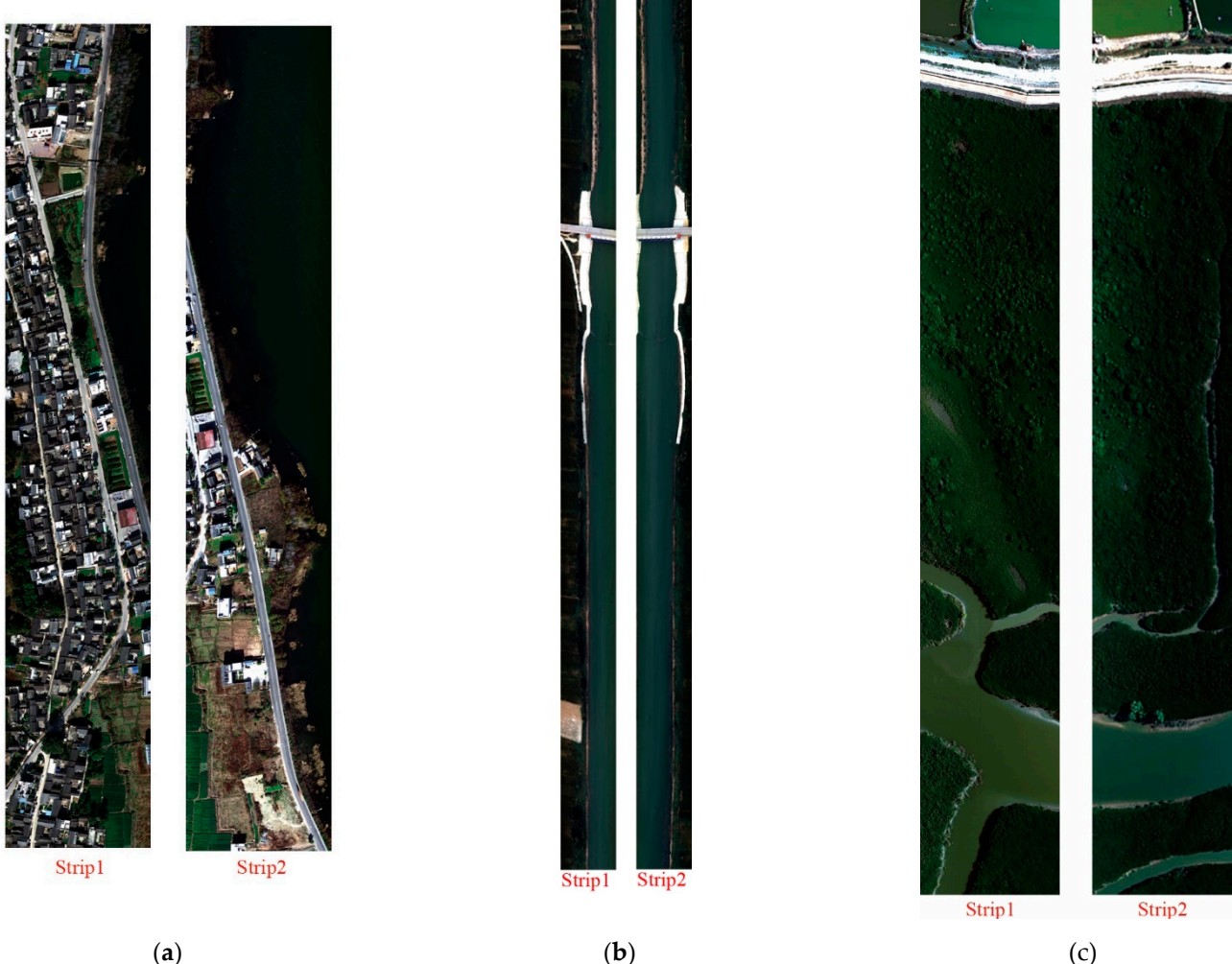

**Figure 3.** True color images of two neighboring image strips (bands of 103, 63 and 27): (**a**) Urban scape; (**b**) River course; (**c**) Forest.

### 2.2. Data Preprocessing

To achieve the image strips that were used in the latter image mosaicking, three preprocessing steps were implemented to the original data set. First, the images that have a large distortion at the beginning and end of each flight line were removed from the original data set. Then, the images that were retained were stitched along the flight line to get the image strips. Finally, the sensor-related calibration and reflectance calculation were implemented for each hyperspectral image strip to convert the digital number to reflectance data. In sensor-related calibration, there were three key steps: spectral calibration, relative radiometric calibration and absolute radiometric calibration [38]. Spectral calibration ensures an accurate instrumental spectral response to surface reflectance, while relative radiometric calibration ensures a uniform output of the sensor across space and time. Absolute radiometric calibration determines the conversion from DN to radiance and to reflectance units. First in spectral calibration, the central wavelength and the full width at half maximum of each spectral band are determined. In radiometric calibration, the corresponding relationship between the DN of each spectral band and the true radiance is established and recorded in the calibration files [19]. Next, the radiance of the pixels in a

hyperspectral image can be calculated using the corresponding calibration files. Finally, the spectral reflectance of each pixel $R_{ij\lambda}$ can be calculated using the following equation:

$$R_{ij\lambda} = \frac{S_{ij\lambda} - D_{ij\lambda}}{T_\lambda - D_{ij\lambda}} \times R_{r\lambda} \tag{1}$$

where $i$, $j$, $\lambda$ represent the pixel row number, column number and wavelength, $S_{ij\lambda}$ is the radiance of the pixel $(i, j)$ in the hyperspectral image, $D_{ij\lambda}$ is the spectral density of dark current, and $T_\lambda$ is the radiance of the reference of a calibration whiteboard in the hyperspectral image which has the reflectance $R_{r\lambda}$.

The reflectance data of the urban scape, river course and forest area are shown in Figure 3a,b. The data size of Urban scape, river course and forest area is 2.6, 6.81 and 3.06 Gigabytes respectively. After the preprocessing of UAV hyperspectral images, the distortion that occurs in the push-broom sensor's scanning procedure along the flight line can be mostly reduced because a gimbal was used in the whole data capture process. However, there are still scale, rotation and translation distortions between the narrow overlapped image strips as shown in Figure 3.

### 2.3. Image Mosaicking Methodology

To get seamless mosaicked image bands, an image mosaicking chain was proposed as shown in Figure 4. Here, the hyperspectral images are the reflectance data calculated from the image strips' DN data, as shown in Figure 3. The geometric rectification (Section 2.3.1) is first implemented. Three kinds of geometric rectification methods based on the GCPs, POS and a combined use of POS and GCPs can be chosen. While geometric errors are unavoidable, the image registration (Section 2.3.2) method was then implemented to adjust the coordinates of the tie points on different image strips to ensure that the neighboring image strips can be stitched together by overlaying with no dislocation. Two image registration methods of the SIFT and RANSAC methods (Section 2.3.2.1) and the improved phase correlation method (Section 2.3.2.2) can be chosen. Finally, the weighted averaging and best stitching image fusion methods can be chosen to mosaic the neighboring image strips into a large seamless image (Section 2.3.3).

### 2.3.1. Geometrical Rectification

Geometric rectification of each strip can reduce the differences and provide the geometric rectified data for image registration. In the POS-based method, the exterior parameters are derived from the POS data, including $(X_s, Y_s)$, which is the location of the light beam center, the pitch, yaw, roller which are the three angles that describe the flight attitude. The focal length $f$ is derived from the camera calibration report. The ground is assumed as a flat plain because the area coverage is small, and then the $Z_A$ coordinate of each ground point $A$ is set as $Z_S$-h; $h$ represents the flight height. The coordinates of each ground point $A$ $(X_A, Y_A)$ is calculated by direct geo-referencing based on the collinear condition equation [39] as shown in Equation (2). In the equation, the $a_1$, $a_2$, $a_3$, $b_1$, $b_2$, $b_3$, $c_1$, $c_2$, $c_3$ are calculated based on the three angles of pitch, yaw, roller to construct the rotation matrix. By a bilinear interpolation method, each image strip can be rectified and resampled to be the coarsely rectified image with geographical coordinates. In this process, the self-developed hyperspetral image processing software, which is programmed by Visual C++, was used.

$$\begin{cases} X_A - X_S = (Z_A - Z_S)\frac{a_1 x + a_2 y - a_3 f}{c_1 x + c_2 y - c_3 f} \\ Y_A - Y_S = (Z_A - Z_S)\frac{b_1 x + b_2 y - b_3 f}{c_1 x + c_2 y - c_3 f} \end{cases} \tag{2}$$

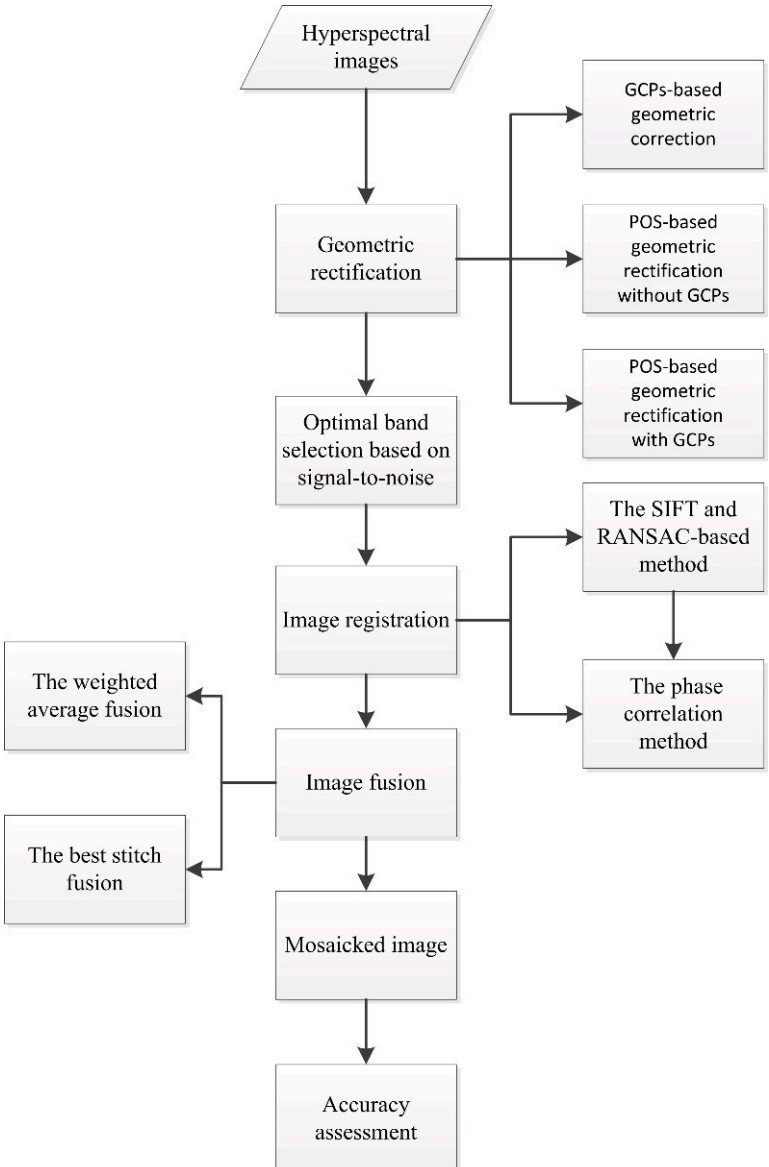

**Figure 4.** The image mosaicking chain.

Due to the limited positioning and attitude accuracy of the POS system on board, there will be an obvious misalignment for the neighboring strips after direct geo-referencing. To improve the accuracy of geometrical rectification, GCP-based image rectification using the curved surface spline function (CSSF) [40] is applied as shown in Equation (3). Each image strip is assigned with new geographical coordinates after the GCP-based image rectification by resampling it with a bilinear interpolation method. As a non-linear transformation function, CSSF only needs three GCPs to simulate a high-order differentiable smooth surface. After the two image strips were geometrically rectified, the smooth surface passes through the same GCPs, so the difference between the pair of rectified image strips becomes smaller. Here, a reference image in the same area was used to extract the GCPs, and ArcGIS 10.1 software was used to implement the CSSF-based image rectification method.

$$W(x,y) = a_0 + a_1 x + a_2 y + \sum_{i=1}^{n} F_i r_i^2 \ln(r_i^2 + \varepsilon) \tag{3}$$

where, $a_0$, $a_1$, $a_2$, $F_i(i = 1, 2, k \ldots n)$ are parameters to be estimated, $r_i^2 = (x - x_i)^2 + (y - y_i)^2$, and $\varepsilon$ is the parameter that is used to adjust the surface curvature. Normally $\varepsilon = 1 \sim 10^{-2}$ when the surfaces are mildly distorted, and $\varepsilon = 10^{-5} \sim 10^{-6}$ when the surfaces are significantly distorted.

### 2.3.2. Image Registration of Neighboring Image Strips

Because noise in the hyperspectral image of one band would result in image matching errors, it is preferred to choose an optimal band that has the least noise and the best image quality to increase the image matching accuracy. Moreover, to make sure that the selected band is suitable for complex scene feature detection, the local variance method based on edge block elimination [41] is used to calculate the image signal-to-noise ratio (SNR) first. Then, the band with the highest SNR was used in the latter image registration.

#### 2.3.2.1. SIFT and RANSAC-Based Method

The traditional feature point-based technical process of image registration is as follows: first the feature points on two images are extracted; then the feature point pairs are matched through feature similarity measurements to obtain tie points; next, the image space coordinate transformation function is established using the tie points; and finally, image registration is performed by coordinate transformation [42].

The image feature point extraction is fundamental for the success of the subsequent image registration. There are many traditional image matching methods, such as scale-invariant feature transform (SIFT) [43] and Speeded-Up Robust Features (SURF) algorithms [42]. The SIFT algorithms run fast and have a high stability in extracting feature point pairs from images with multiple changes such as translation, rotation, and zooming scale. In using a SIFT algorithm, there exist many image matching strategies such as the brute-force matching [44], the K-nearest neighbor matching [45] and the best-bin-first matching methods [46]. For simplicity, the nearest neighbor ratio method [47] is used in image matching first. However, there are some inevitably wrong matched points. So the random sample consensus (RANSAC) algorithm, which is effective and fast in removing some outliers and mismatched points are used for purifying the tie points [48]. After extracting the tie points, the spatial similarity transformation [39] with 7 parameters (3 rotation parameters, 3 translation parameters, 1 scale parameter) was applied to the right image strip with the left image strip as the reference image.

#### 2.3.2.2. The Improved Phase Correlation Method

With consideration that the phase correlation method [49] based on the two-dimensional Fourier transform is scene-independent, and can accurately align two-dimensional images with small translation, an improved phase correlation method was proposed and compared with the SIFT and RANSAC methods. The advantages of this method include its simplicity, its fast running speed, its insensitivity to brightness changes, and its strong anti-interference ability.

First, to deal with the image strips with rotation and scaling, the phase correlation method was improved by transforming the images from the Cartesian coordinate system to the polar coordinate system first, then the rotation relationship between the images is represented as $\theta' = \theta + \theta_0$, where $\theta$ represents the scaling relationship represented as $r' = r_0 r$ and transferred to an additive relationship by logarithm operation [50].

The main steps of the phase correlation method accounting for the rotation and scaling differences are:

(1) Transfer the image from the Cartesian coordinate system to the polar coordinate system.

In the Cartesian coordinate system, suppose the relative rotation angle is $\theta_0$ and scaling factor is $k$, then the spatial relationship between the neighboring image strip can be represented in Equation (4):

$$f_2(x,y) = f_1[k(x\cos\theta_0 + y\sin\theta_0), k(-x\sin\theta_0 + y\cos\theta_0)] \tag{4}$$

(2) Then the Fourier transformer of the images $F_1$ and $F_2$ satisfies Equation (5):

$$F_2(u,v) = \frac{1}{k^2}e^{-i2\pi(ux_0+vy_0)}F_1\left[\frac{1}{k}(u\cos\theta_0 + v\sin\theta_0), \frac{1}{k}(-u\sin\theta_0 + v\cos\theta_0)\right] \tag{5}$$

(3) Next, the amplitude spectra of the two images have the following relationship:

$$M_2(u,v) = \frac{1}{k^2}M_1\left[\frac{1}{k}(u\cos\theta_0 + v\sin\theta_0), \frac{1}{k}(-u\sin\theta_0 + v\cos\theta_0)\right] \tag{6}$$

Set $\begin{cases} u = \rho\cos\theta \\ v = \rho\sin\theta \end{cases}$, Equation (6) can be simplified as:

$$M_2(\rho\cos\theta, \rho\sin\theta) = \frac{1}{k^2}M_1\left[\frac{\rho}{k}\cos(\theta-\theta_0), \frac{\rho}{k}\sin(\theta-\theta_0)\right] \tag{7}$$

(4) Based on Equation (7), the parameter $\frac{\rho}{k}$ can be transformed by the logarithmic transformation $\lg\frac{\rho}{k} = \lg\rho - \lg k$, then Equation (7) can be transformed to Equation (8):

$$M_2(\theta, \lg\rho) = \frac{1}{k^2}M_1[(\theta-\theta_0), \lg\rho - \lg k] \tag{8}$$

(5) Using Equation (8), the phase correlation method can be used to calculate the rotation angle $\theta_0$ and scaling parameter $k$.

Secondly, considering that in fast Fourier transform, if the image size $N$ is the power of 2 ($N = 2^p$, $p$ is an integer), its calculation efficiency is highest, sub-images (Figure 5), which are cut from the original image with the image size of 2-power, can be used to calculate the transformation parameters first. Then, the translation offset for the original images can be calculated according to the location of sub-images. There are two overlap patterns including the left–right overlap as shown in Figure 5a and the up-down overlap as shown in Figure 5b. Using the 2-power sub-image, the image size becomes smaller, the overlapped rate is increased and the efficiency can be promoted.

Thirdly, when the image has many distortions or noises, the image registration result may be inaccurate [51]. Since image edges are the most basic and relatively stable feature, edges can be used to promote the efficiency and accuracy of the phase correlation method.

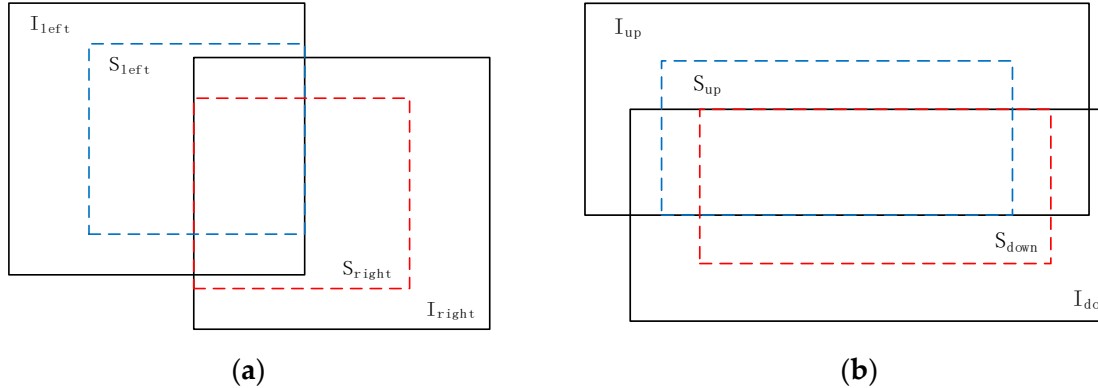

(a)   (b)

**Figure 5.** Two-power sub-images (the blue and red line represent the boundaries of 2-power sub-images): (**a**) the left-right overlap 2-power sub-images; (**b**) the up-down overlap 2-power sub-images.

The steps of the improved phase correlation (ICP) method includes:

1. The 2-power sub-images of the reference image $F_1(x, y)$ and the registered image $F_2(x, y)$ are set as $f_1(x, y)$ and $f_2(x, y)$ respectively. The Canny edge detector is used to extract the edge of $f_1(x, y)$ and $f_2(x, y)$, which are represented as $f_1'(x, y)$ and $f_2'(x, y)$.

2. The Fourier transformation is implemented to $f_1'(x, y)$ and $f_2'(x, y)$ respectively, and then the amplitude spectra are obtained as $M_1(u, v)$ and $M_2(u, v)$. The amplitude spectra is converted from the Cartesian coordinate system to the polar coordinate system, and $M_1(\theta, \lg\rho)$ and $M_2(\theta, \lg\rho)$ were got by logarithmic operation.

3. The traditional phase correlation method is applied to $M_1(\theta, \lg\rho)$ and $M_2(\theta, \lg\rho)$ to get the rotation angle $\theta_0$ and the scale parameter $k$. Then, the $\theta_0$ and $k$ are used to do the inverse transformation to the 2-power sub-image of the registered image $f_2(x, y)$ to get a transition image $f_3(x, y)$ with only translation offsets.

4. Canny edge detector is used to get the edge image of $f_3(x, y)$ as $f_3'(x, y)$. Then, the phase correction method is used to calculate the relative translation offset $(x_0, y_0)$ between the registered image $F_2(x, y)$ and $f_3(x, y)$.

5. Based on the calculated $\theta_0$, $k$ and $(x_0, y_0)$, the spatial similarity transformation is used and incorporated in a bilinear interpolation resampling procedure is used in image registration to produce the geometrically rectified hyperspectral image strips for the latter seamless mosaicking.

### 2.3.3. Image Fusion

Hyperspectral images are wete used for water monitoring, forestry analysis, mineral exploration, etc. For these applications, spectra fidelity of mosaicked images must be ensured. To eliminate the stitching line [52], the weighted average [53] and the best stitch fusion methods [54] were used. To analyze the spectral fidelity of the mosaicked image, the spectral reflectance curves of the different types of ground objects' typical pixel in the overlapped area on the original, the geometrically corrected and the mosaicked images were compared. The spectral angle cosine (SAC), spectral correlation (SC), spectral information divergence (SID) and Euclidean distance (ED) [55] were used to measure the similarity of spectral curves in the overlapped area before and after image fusion. The greater the SAC value and the SC value, the greater the spectral similarity is, and the smaller the SID value and the ED value, the greater the spectral similarity is.

## 3. Experimental Results

As shown in Figure 4, different methods can be chosen in geometrical rectification, image registration and image fusion. In the following, the experiment results of different methods are presented. First, because few distinct GCPs can be extracted in the reference image in the river course area, the river course area is chosen as the representative study area to show the results of the different geometric rectification methods with or without GCPs. Secondly, the results of the two image registration methods of all the three study areas are presented. Thirdly, because the urban scape area consists of many types of ground object with different spectral feature characteristics, the two image fusion strategies are compared using the urban scape area as the representative study area. Moreover, the weighted average fusion results of the river course and forest areas are presented.

### 3.1. Geometric Rectification Results of River Course Area

Because a large portion of the river course area is dominated by water, the ground control points were derived from google earth (GE) images rather than from field surveys in the research. Eleven GCPs shown in Figure 6a were used for geometrical rectification.

Ten other GCPs in the GE image were used to assess the geometrical accuracy. As shown in Table 3, geometric rectification with POS data and GCPs is a compromise between efficiency and accuracy, and improves the accuracy compared to geometric rectification using only POS data. The medium error of the rectified image was 1.76 m, while the spatial

resolution of the GE image was 2.5 m. The geometrical rectification error was less than 1 pixel, in accordance with the reference image.

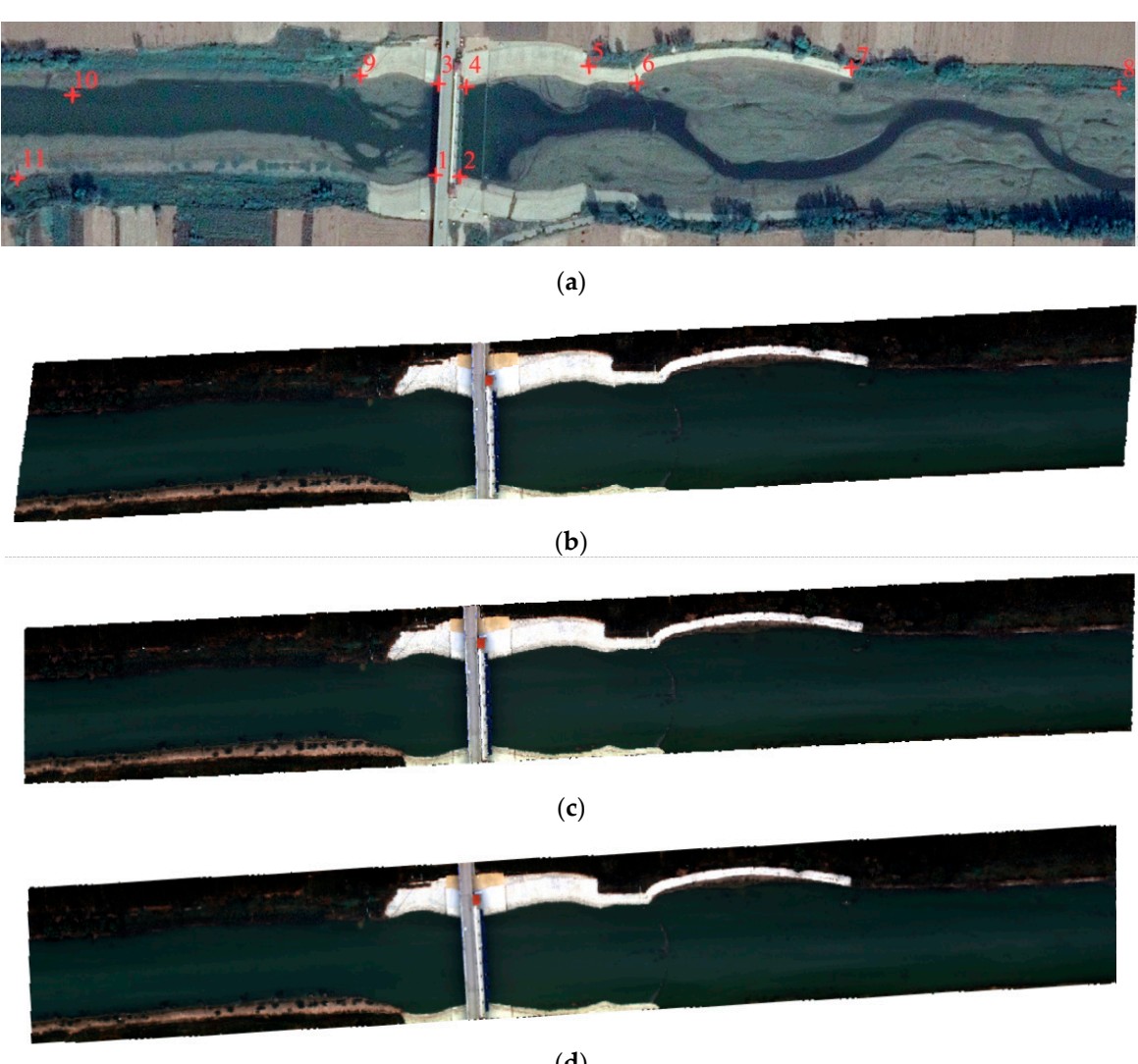

(**a**)

(**b**)

(**c**)

(**d**)

**Figure 6.** River course GCPs and geometric rectified results: (**a**) GCPs on GE image; (**b**) CSSF method with 11 GCPs; (**c**) POS-based method without GCP; (**d**) POS-based geometric rectification method with 4 GCPs that labeled as 2, 6, 8 and 11 on Figure 5a.

**Table 3.** Geometric rectification accuracy (units in meters) of the river course area.

|  |  | Mean Residual | Medium Error | Mean Absolute Deviation | Median Absolute Deviation | Standard Deviation | Maximum Residual |
|---|---|---|---|---|---|---|---|
| CSSF method with 11 GCPs | $x$ | 0.0862 | 0.6954 | 0.6068 | 0.8621 | 0.6900 | 1.0909 |
|  | $y$ | −0.1198 | 1.6207 | 1.2474 | 2.2333 | 1.6163 | 2.8232 |
|  | $xoy$ |  | 1.7636 |  |  | 1.7574 |  |
| POS-based method without GCPs | $x$ | −4.1354 | 4.3593 | 1.1555 | 1.1744 | 1.3792 | 6.8187 |
|  | $y$ | −0.0961 | 2.2469 | 2.1094 | 1.9082 | 2.2449 | 3.8767 |
|  | $xoy$ |  | 4.9043 |  |  | 2.6347 |  |
| POS-based method with 4 GCPs | $x$ | −1.2553 | 1.7413 | 1.026 | 0.5031 | 1.2068 | 3.372 |
|  | $y$ | 0.2162 | 1.6208 | 1.3627 | 1.0367 | 1.6063 | 3.1469 |
|  | $xoy$ |  | 2.3789 |  |  | 2.0091 |  |

*3.2. Image Registration Results*

As shown in Figures 7 and 8, the ICP image registration method can get comparative results in the urban scape and river course areas; however, it failed in the forest area.

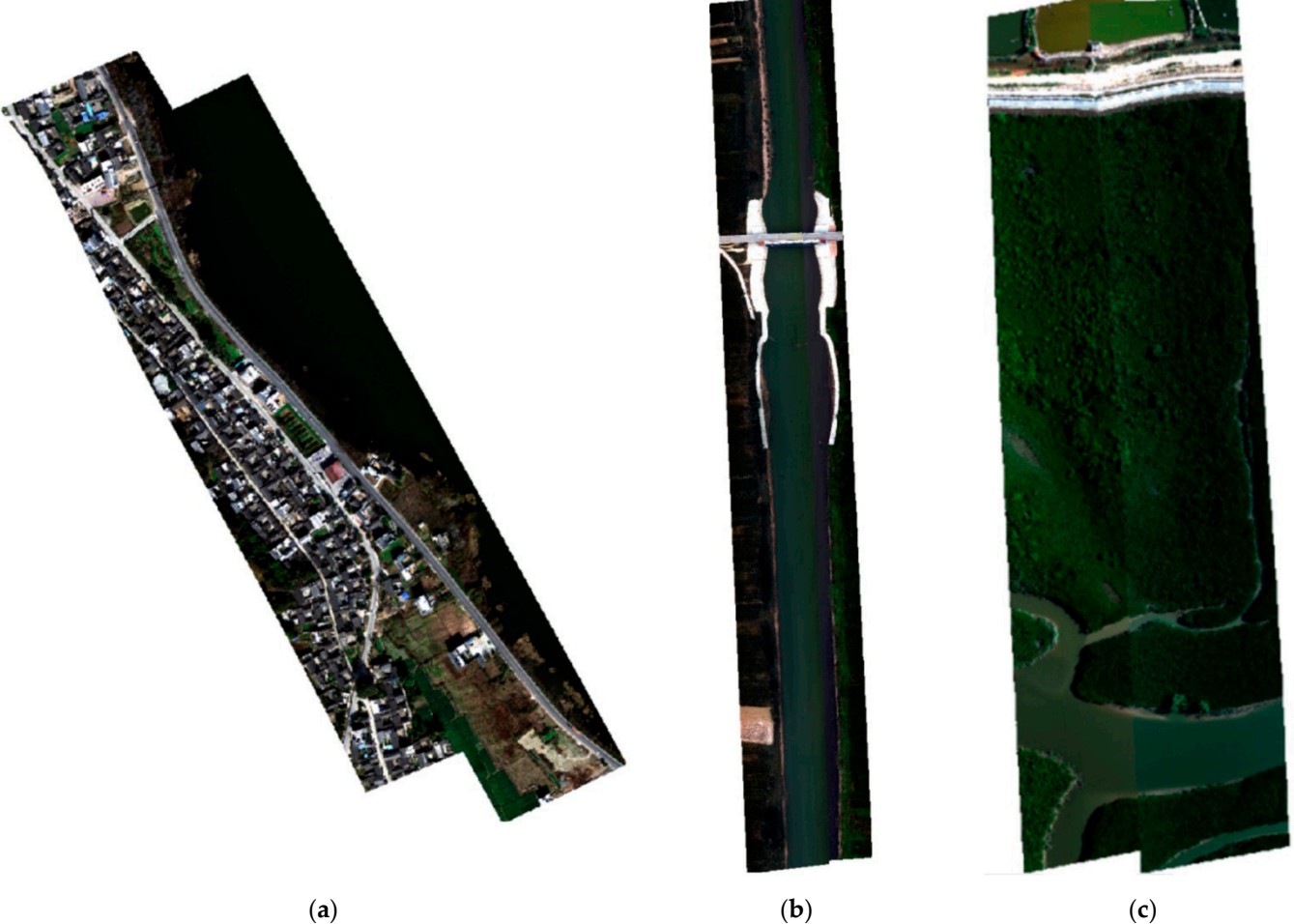

(**a**) (**b**) (**c**)

**Figure 7.** SIFT and RANSAC-based image registration results: (**a**) Urban scape; (**b**) River course; (**c**) Forest.

To compare the image registration methods, the testing points derived from the GE image for image rectification accuracy assessment were used to evaluate the accuracy for each study area after image registration. The time cost and accuracy are first analyzed for the river course and urban scape areas. The computer setting is as follows: Intel(R) Xeon (R) CPU E5-2630 v4 @ 2.20 GHz 2.20 GHz, the memory storage is 32G.

As shown in Tables 4 and 5, for the urban scape and river course areas, the SIFT and RANSAC methods cost more time to establish the transformation model with a similar accuracy. The IPC method first extracts the edge image from the initial rectified image's 2-power sub-image, and then used Fourier transformation to convert the edge image from the spatial space to the frequency space; the transform parameters are calculated by the traditional phase correlation method, and finally the transform parameters of the original images are calculated. The time cost is much less than the traditional SIFT- and RANSAC-based methods.

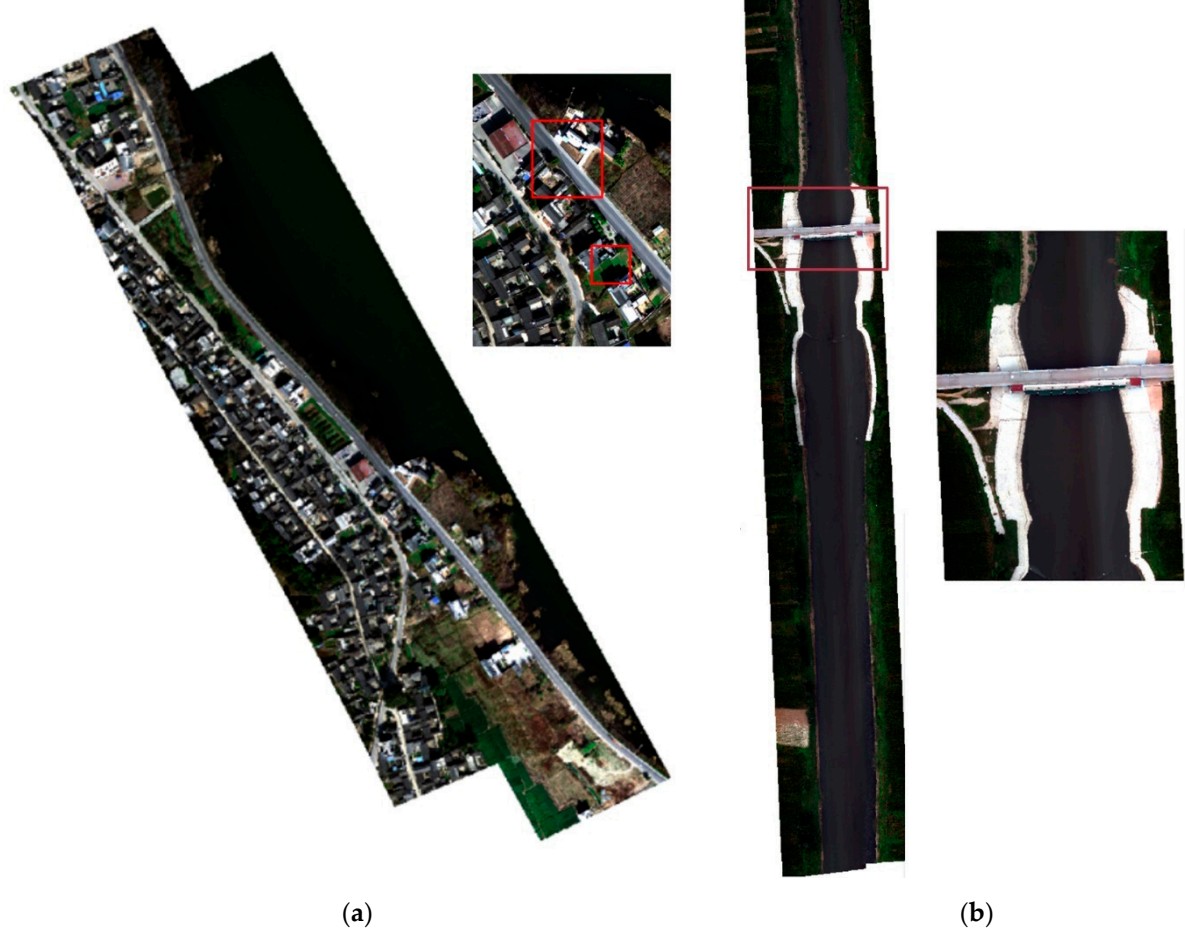

**Figure 8.** Image registration results using the improved phase correlation method: (**a**) Urban scape; (**b**) River course.

**Table 4.** The image registration accuracy—urban scape (units in meters).

| | | Mean Residual | Medium Error | Mean Absolute Deviation | Median Absolute Deviation | Standard Deviation | Maximum Residual | Time Cost |
|---|---|---|---|---|---|---|---|---|
| The improved phase correlation method | $x$ | −0.67025 | 1.1664 | 1.5992 | 0.6899 | 1.1019 | 3.5498 | |
| | $y$ | 0.24754 | 1.0813 | 1.307 | 1.5517 | 0.8618 | 3.4393 | 0.329s |
| | $xoy$ | | 1.5905 | | | 1.3989 | | |
| The SIFT and RANSAC-based method | $x$ | −1.6731 | 1.2104 | 0.95694 | 2.8478 | 1.1326 | 3.0629 | |
| | $y$ | −0.82138 | 1.1552 | 0.97138 | 1.0708 | 0.8137 | 2.7707 | 5.395s |
| | $xoy$ | | 1.6731 | | | 1.3945 | | |

**Table 5.** The image registration accuracy—river course (units in meters).

| | | Mean Residual | Medium Error | Mean Absolute Deviation | Median Absolute Deviation | Standard Deviation | Maximum Residual | Time Cost |
|---|---|---|---|---|---|---|---|---|
| The improved phase correlation method | $x$ | −2.7975 | 1.4856 | 1.7581 | 0.95615 | 1.6069 | 3.7709 | |
| | $y$ | −0.95435 | 1.2470 | 0.73139 | 0.41235 | 0.95064 | 2.8895 | 1.169s |
| | $xoy$ | | 1.9395 | | | 1.8670 | | |
| The SIFT and RANSAC-based method | $x$ | −1.1530 | 1.2191 | 1.7520 | 1.0180 | 1.5961 | 4.2247 | |
| | $y$ | −0.97421 | 1.2613 | 0.63685 | 0.19585 | 0.80109 | 2.5976 | 4.082s |
| | $xoy$ | | 1.7541 | | | 1.7858 | | |

For the forest data, the SIFT and RANSAC-based methods get good image registration results, with the medium error of 1.9669m, less than one pixel of the GE image, as shown in Table 6. However, the ICP method failed, with no forest result in Figure 8. For the spectral complex environment such as forest, the high spatial and spectral resolution image was characterized by the spectral heterogeneity and texture, and led to the difficulty of extracting accurate edges from the images, so the IPC method cannot get the accurate spatial similarity transformation parameters that are needed to co-register the neighboring image strips.

**Table 6.** The SIFT and RANSAC-based image registration accuracy—forest (units in meters).

|  | Mean Residual | Medium Error | Mean Absolute Deviation | Median Absolute Deviation | Standard Deviation | Maximum Residual |
|---|---|---|---|---|---|---|
| $x$ | 0.0864 | 0.9065 | 0.7977 | 1.4019 | 0.9024 | 1.3841 |
| $y$ | 0.2166 | 1.7455 | 1.3750 | 2.6205 | 1.7321 | 3.0199 |
| $xoy$ |  | 1.9669 |  |  | 1.9530 |  |

### 3.3. Image Fusion Results

As shown in Figures 7 and 8, the spectral differences between the left and right geo-registered images are obvious. To get seamless image mosaicking results, the weighted average and the best stitching image fusion methods were used and compared in experiments.

#### 3.3.1. Urban Scape Image Fusion Results

The weighted average fusion result and the best stitching fusion result of the urban scape areas are shown in Figures 9 and 10. As shown in Figure 9, for images with complex features, the weighted average fusion is not suitable, and the red house has obvious ghosting problems. Since the best stitching line fusion takes into account the color and structure of the image, the stitching line avoids the ground objects with large differences in color and geometric structure, such as passing the houses, so the ghosting problem can be avoided and a better fusion effect is achieved for the urban scape area in Figure 10.

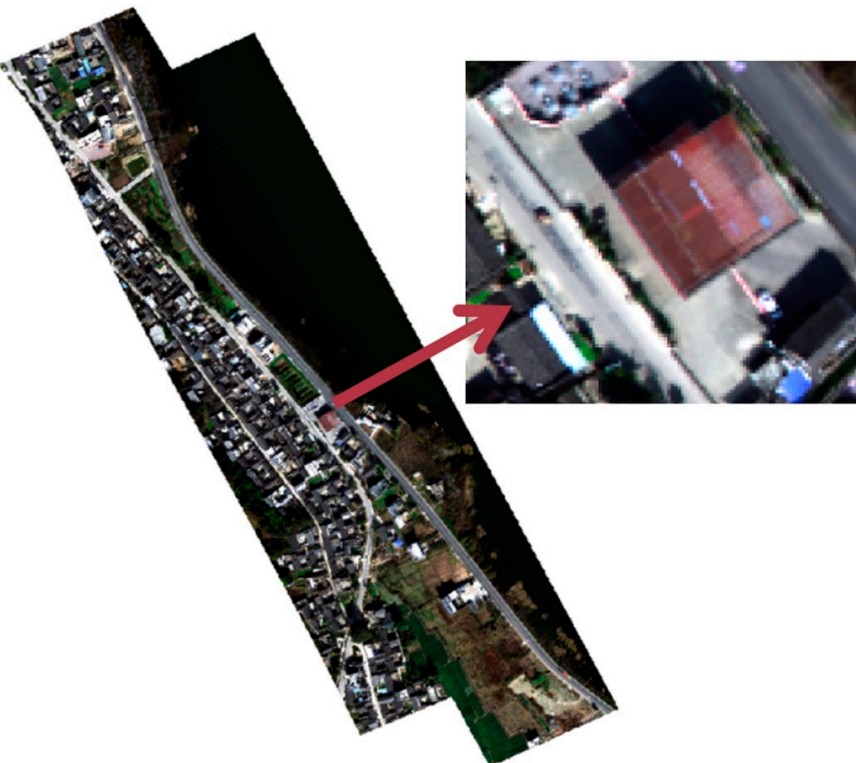

**Figure 9.** The weighted average fusion result of a urban scape area.

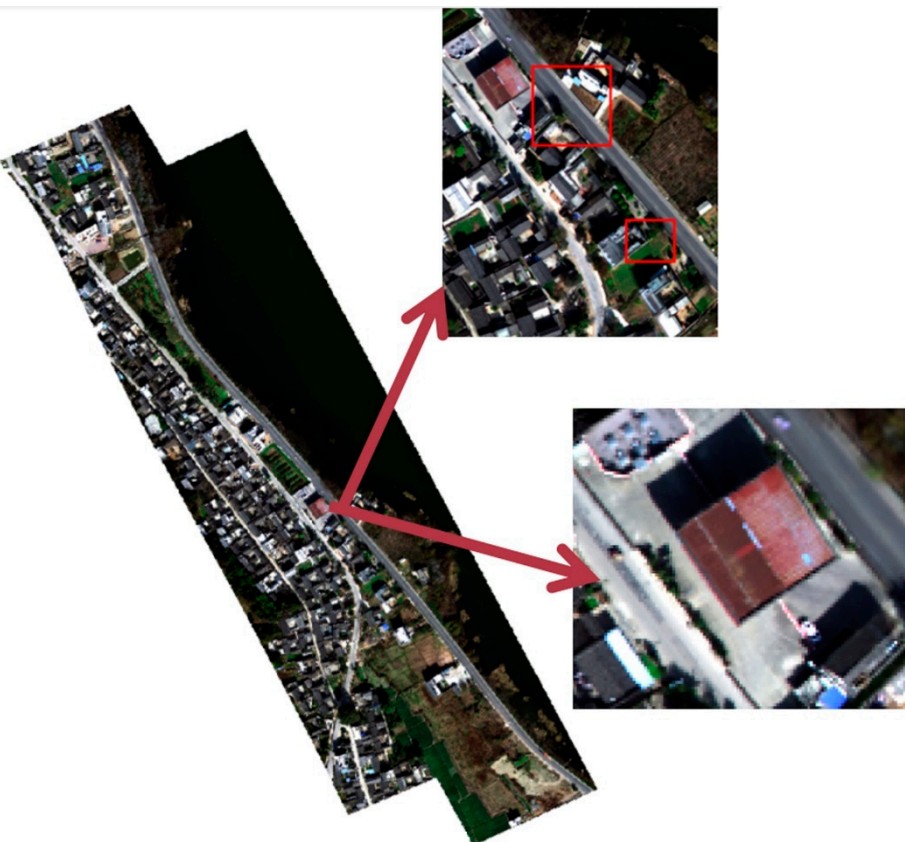

**Figure 10.** The best stitching image fusion result of an urban scape area.

To further evaluate the best stitching fusion method for the urban scape area, the five typical ground objects including buildings, water, soil, farmland and roads in the overlapped area between the left and right image strips are compared with their spectral curves between the original, geo-registered and the mosaicked hyperspectral images (Figure 11). The spectral curve was extracted from one typical pixel of each type of ground object in the overlapped area between the neighboring image strips. From Figure 11, it can be seen that the difference between the spectra of the left and right original image strips is obvious especially for soil, farmland and road ground objects. These are mainly due to the following reasons.

First, the illumination difference of the round-trip flight strips; secondly, only one white board is set in the experimental area, and there may exist errors in the radiation calibration of images. Moreover, as the original images are re-sampled after the geometric rectification and image registration, the change in the original spectrum makes the left and right rectified images have larger spectral reflectance differences, especially for soil (600–700 nm), with a maximum difference of about 0.125, farmland (750–850 nm) with a maximum difference about 0.008, and road (550–850 nm) with a maximum difference about 0.03. After image fusion, the spectral curves of different ground objects (red curves) on the mosaic image is closer to that on the left registered (Rec_left) image (green curves) so the stitching line will disappear after image fusion, as shown in Figure 10. To find out whether the spectral reflectance before and after image mosaicking changes a lot, the spectral similarity assessment indices were calculated between the spectral curve of the mosaicked image and the original left and right images and the Rec_left image and Rec_right image as shown in Table 7. As can be seen from Table 7, the typical pixel of each type of ground object extracted from the mosaicked image has similar spectral reflectance to the Original left and Rec_ left images, which indicates that the spectral similarity of the pixels of the overlapped area before and after the mosaicking is high. Image fusion can reduce the spectral difference of the overlapped area's ground object.

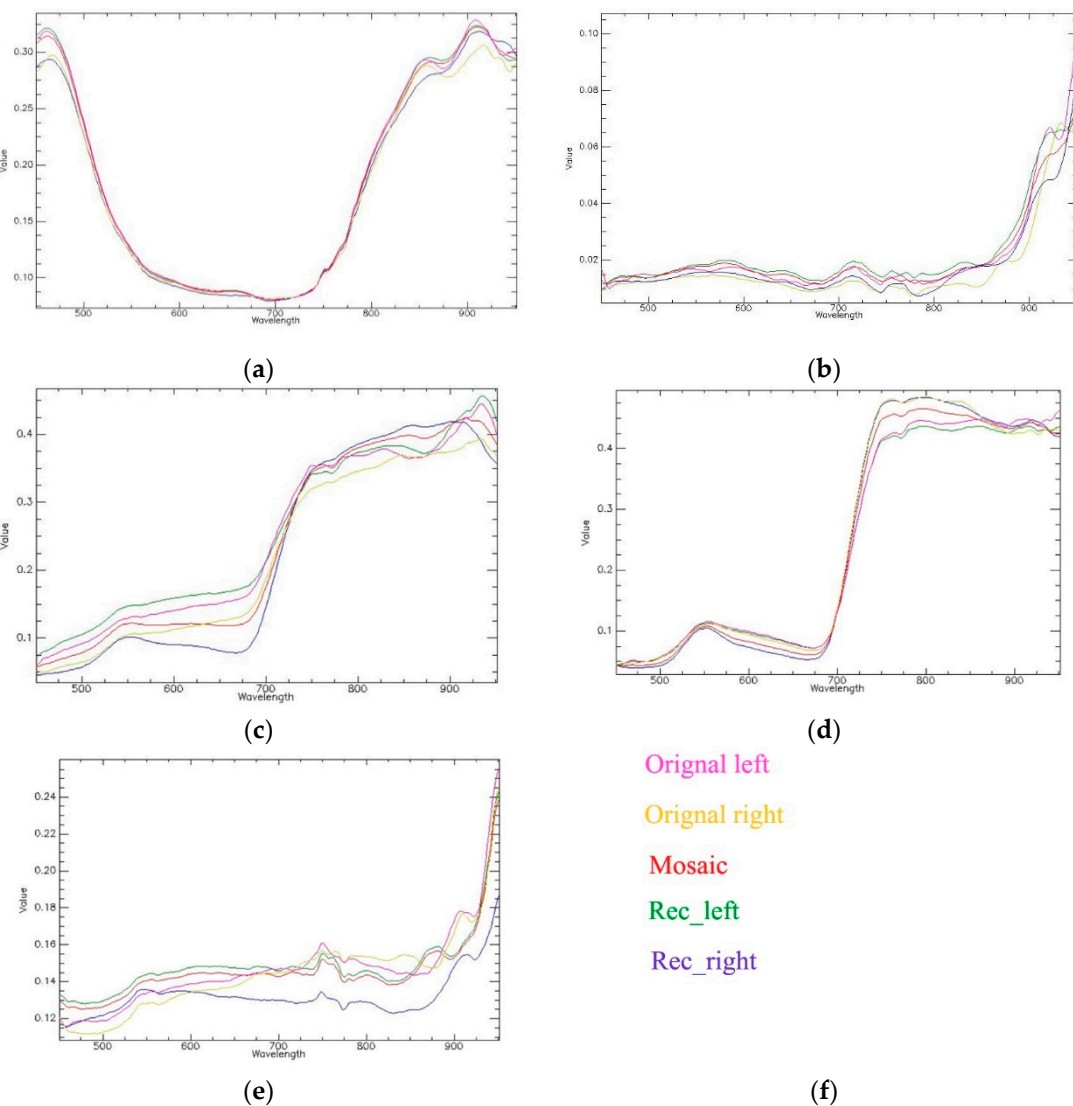

**Figure 11.** Comparison of typical spectral curves before and after image mosaicking for the urban scape area: the spectral curves of objects of (**a**) building; (**b**) water; (**c**) soil; (**d**) farmland; (**e**) road are represented in five colors, the sources of the spectral curves are shown in (**f**) using the texts in the corresponding colors, where the original_left means the original left image, original_right means the original right image; mosaic means the mosaicked image; Rec_left means the geo-registered left image, Rec_right means the geo-registered right image.

**Table 7.** Evaluation of similarity of spectral curves of different ground objects in the overlapped area before and after image mosaicking (urban scape).

| Ground Objects | The Compared Image | SAC | SC | SID | ED |
|---|---|---|---|---|---|
| building | Rec_left image | 0.9999 | 0.9998 | 0.00002 | 0.0377 |
| | Rec_right image | 0.9997 | 0.9984 | 0.00022 | 0.1114 |
| | Original left image | 0.9999 | 0.9997 | 0.00004 | 0.0407 |
| | Original right image | 0.9998 | 0.9990 | 0.00023 | 0.1313 |
| water | Rec_left image | 0.9989 | 0.9964 | 0.00095 | 0.03670 |
| | Rec_right image | 0.9934 | 0.9863 | 0.00479 | 0.05112 |
| | Original left image | 0.9923 | 0.9894 | 0.00548 | 0.05837 |
| | Original right image | 0.9857 | 0.9776 | 0.01066 | 0.07115 |

**Table 7.** *Cont.*

| Ground Objects | The Compared Image | SAC | SC | SID | ED |
|---|---|---|---|---|---|
| soil | Rec_left image | 0.9951 | 0.9921 | 0.00396 | 0.3190 |
| | Rec_right image | 0.9968 | 0.9969 | 0.00670 | 0.3277 |
| | Original left image | 0.9971 | 0.9930 | 0.00745 | 0.4289 |
| | Original right image | 0.9992 | 0.9968 | 0.00143 | 0.3315 |
| farmland | Rec_left image | 0.9991 | 0.9984 | 0.00209 | 0.1664 |
| | Rec_right image | 0.9997 | 0.9995 | 0.00071 | 0.1927 |
| | Original left image | 0.9986 | 0.9973 | 0.00231 | 0.2403 |
| | Original right image | 0.9995 | 0.9987 | 0.00087 | 0.2628 |
| road | Rec_left image | 0.9999 | 0.9992 | 0.00001 | 0.0457 |
| | Rec_right image | 0.9986 | 0.9111 | 0.00105 | 0.1207 |
| | Original left image | 0.9986 | 0.9574 | 0.00122 | 0.2362 |
| | Original right image | 0.9976 | 0.9091 | 0.00217 | 0.1511 |

### 3.3.2. The Weighted Average Fusion Results of River Course and Forest Area

To further evaluate the image fusion methods, the river course and forest images before and after the weighted average fusion are shown in Figures 12 and 13. As shown in the red rectangle on Figures 12 and 13a, the spectral differences that are obvious in Figures 7c and 8b disappear. Figure 13b shows the mosaicked image of multiple image strips of the mangrove forest. The mosaicked images have a good spectral smoothing quality in the overlapped area. It can be seen that the weighted average fusion method can get good results for the spectral homogeneous environments, such as river course and forest areas. With the consideration that the weighted average fusion method is more convenient than the best stitching fusion method, it is suggested to use the weighted average fusion method for the water and vegetation-dominated areas when the data volume is huge. However, when the computation ability is essential, the best stitching fusion method can perform better than the weighted average fusion method.

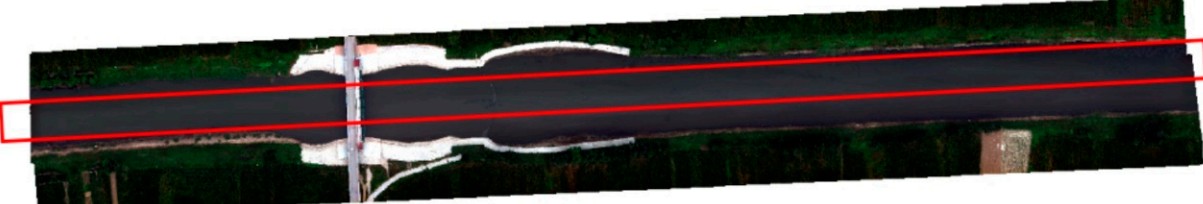

**Figure 12.** The weighted average fusion result of river course area.

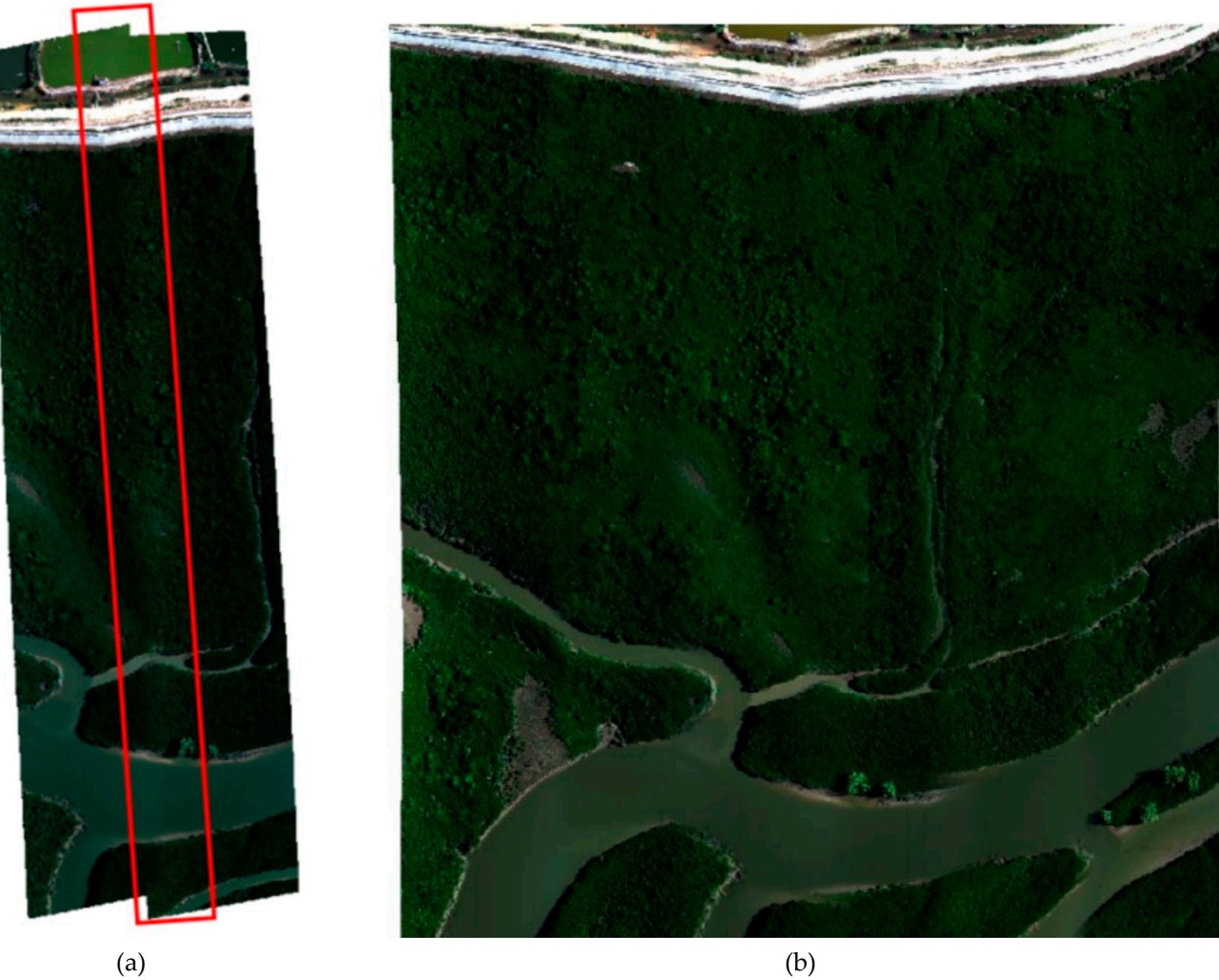

(a)                                             (b)

**Figure 13.** Forest experiment results: (**a**) the fused image by the weighted average method; and (**b**) the fused image of multiple forest image strips.

## 4. Discussion

UAV-based image products are generally produced by stitching together hundreds of partially overlapping image strips or frames captured in certain geometrical patterns [56]. However, when "matching" any two images, the transformation and reprojection using various algorithms routinely introduce localized distortions. While the positional accuracy of an individual image may be on the order of a few centimeters, the accuracy of the completed mosaic may increase to the decimeter range as the positional errors accumulate through the merging process. Accurate overlapping of swaths requires a good number of matching GCPs to avoid further distortion in the final mosaic [30]. To improve the geometrical accuracy, some researchers used UAV orthomaps as the reference data in image mosaicking. The identification of tie points in high resolution imagery, especially those exhibiting different geometric and spectral characteristics, is mainly established by applying a feature-based approach such as the modified SURF algorithm [28]. This method needs an accurate reference image in advance and much computational support for improving the automation and efficiency of its processing procedures [17]. Some other researchers used multispectral UAV imagery to obtain Ortho-mosaicked NDVI images for precision agriculture [57]. They used a common RGB camera and a modified camera sensitive to near-infrared radiation to gather images. The OpenDroneMap (ODM) was used to merge the two sets of images into ortho-mosaics. This work provided an efficient

way to acquire large-area data for agriculture research using UAV. This method used the common aerial photogrammetry technique that required high image overlapping rates.

In comparison, our work is focused on the push-broom hyperspectral image mosaicking between image strips. The proposed chain of image mosaicking mainly includes three steps of geometrical rectification, image registration and image fusion.

### 4.1. Geometric Rectification

For the study areas that are dominated by water or forest there doesn't exist so many distinct feature points that can be easily collected as GCPs. The traditional image matching approaches cannot extract sufficient tie points. To improve the efficiency and accuracy of geometrical correction, the different strategies of image rectification and image registration were compared in experiments. First, as shown in the image rectification experiments of the river course, the medium error of the POS-based method is 4.9 m, POS alone achieved a low quality of UAV georeferencing, especially in push-broom spectrometers [18]. Navigation grade GPS sensors do not facilitate hyperspectral push-broom scanning on a UAV [58]. The use of ground control points (GCPs) is absolutely necessary for improving georeferencing. Recent advances in direct georeferencing and imaging technologies—like the GNSS and INS—enable precise mapping using a minimal number of GCPs [30]. The medium error of the CSSF method with GCPs is 1.76 m, while the medium error of the POS-based method with less GCPs is 2.37 m. Geometric rectification based on the combination of POS data and GCPs is a compromise between efficiency and accuracy, and it not only reduces the demand for a large number of GCPs, but also improves the accuracy compared to geometric rectification using only POS data. In the experiments, the CSSF method is used to calculate the coordinates of rectified images with GCPs. By setting appropriate parameters, the modeled curved surface passes through each known GCP on the rectified images, and it is useful in eliminating the local deformation of images around GCPs and will benefit subsequent image registration. In all, the geometric correction accuracy of the mosaicked image depends mainly on the accuracy and the number of GCPs. In the study, GCPs were manually extracted from the GE images, so the GCP location errors may impact the geometric rectification of hyperspectral images, and the medium errors of geo-rectification results are almost 2 m for different study areas. In future, with more precise GCPs and as the amount of GCPs becomes larger, the geometric accuracy of the mosaic will increase.

### 4.2. Image Registration

To select an optimal band from the hyperspectral image before image registration, the local variance method based on edge block elimination [41] was used to calculate SNR for data quality characterization assessment of each band with high efficiency and precision [59]. In the image registration, the band with the highest SNR was selected in calculating the transform parameters between neighboring images. The SIFT and RANSAC-based method and the improved phase correlation method were compared from the aspects of time cost and geometric accuracy.

In image registration, the traditional phase correlation method is improved by using a 2-power sub-image and the edge information in the calculation of transformation parameters. The image registration experiment results of three types of ground areas show that the improved phase correlation method can achieve comparable results to the SIFT and RANSAC-based image registration method for study areas with distinct features such as urban scape and river course areas with less time cost. However, for the forest area, the improved phase correlation method failed and the SIFT and RANSAC-based method is suitable. To improve the effectiveness of the improved phase correlation method in areas with complex textures such as forest, further work needs to be done to find ways to calculate transformation parameters correctly.

*4.3. Image Fusion*

Another contribution of our work is that the spectral fidelity of each kind of ground object on the image before and after image mosaicking is ensured. The two image fusion methods were compared by both qualitative and quantitative assessments. In study areas such as the forest and the river course, the spectra of ground objects in the overlapped areas of images are almost homogeneous, the weighted average fusion method is suggested due to its simplicity for calculation. However, the urban scape area consists of various land cover types, and its spectra are variable in the overlapped area. The best stitching fusion method is suggested because it can avoid the ghosting problem that occurs when the weighted average fusion method was used.

**5. Conclusions**

This paper proposed a simple and effective chain of methods to produce geometrically corrected and mosaicked images with high spectral fidelity from UAV push-broom hyperspectral images. The proposed workflow generates a mosaicked output progressively by three major steps, including image rectification, registration and fusion. In the image mosaicking, the design of each step should be considered according to the application purpose. In our research work, the spectral homogenous area such as the river course and forest area always needs GCPs to get higher geometric accuracy. However, in practice, it is hard to get GCPs in such places by a field survey. Using Google earth images or another source of images as reference data, the proposed image mosaicking chain integrates the POS information, GCPs and the tie points together to refine the geometric accuracy of the mosaic. Researchers that are interested in using the UAV-based push-broom images and multi-source images together in environment monitoring can implement the proposed methodology to get a co-registered image data set. The image mosaic is of a high spectral fidelity. While it is essential to acquire data that can preserve scientific rigor, the radiometric calibration of the mosaic is beyond the scope of our research. There still remains some further works that can be done in the future:

(1) In the geometric rectification, the influence of the number and distribution of GCPs on the correction accuracy needs to be further studied. Since GCPs also have inherent errors, how to develop a method that can take into account the uncertainty of GCPs remains an issue that needs further investigation. In POS-based geometric rectification, this paper assumes that the scene area is a flat plain, which is not true for some study areas. Methods are yet to be developed to apply POS-based geometric rectification to non-flat terrains.

(2) In selecting the optimal band for image registration, SNR is used. However, the band with a higher SNR may have less information. This will also affect the accuracy of subsequent image registration and the subsequent processing steps. Therefore, other band selection methods may be explored to find an optimal band with both a high SNR and a high information content.

(3) Differences in reflectance between tie points in the original image strips lead to stitching lines. We mainly studied image fusion methods to eliminate the spectral difference in the overlapped area between the image strips, but did not calibrate the spectral reflectance of hyperspectral images against ground-based spectral measurements.

(4) The number of bands of the hyperspectral images used in this study is large, and with the continuous improvement in image resolution, the amount of data will become even larger in the future. It is necessary to study fast mosaicking methods or use multiple parallel computing systems to improve the efficiency of data processing.

**Author Contributions:** Conceptualization, L.Y.; methodology, G.Z.; validation, X.X.; data resources, X.M. and W.G.; data curation X.X.; writing—original draft preparation, L.Y.; writing—review and editing, J.M.C. All authors have read and agreed to the published version of the manuscript.

**Funding:** This research was funded in part by the State Scholarship Fund of China (No. 201906435005), Chinese Academy of Sciences Strategic Leading Science and Technology Project (Class A) (XDA13020506), National Natural Science Foundation of China (61405204), China's National Key R&D Programme

(2018YFB0504903, 2016YFB0501402), the High Resolution Remote Sensing, Surveying and Mapping Application Demonstration System (Phase II) (42-Y30B04-9001-19/21), the Key Program of Marine Economy Development Special Foundation of the Department of Natural Resources of Guangdong Province (GDNRC [2020]012) and in part by the Key Deployment of Chinese Academy of Sciences (Y9F0600Z2F) and by the Fundamental Research Funds for the Central Universities(2021YQDC06) of China university of mining and technology-Beijing.

**Informed Consent Statement:** Not applicable.

**Data Availability Statement:** Not applicable.

**Conflicts of Interest:** The authors declare no conflict of interest.

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
