# Peer review of "Seamless Mosaicking of UAV-Based Push-Broom Hyperspectral Images for Environment Monitoring"

_remotesensing, doi:10.3390/rs13224720_

Round 1

Reviewer 1 Report

The manuscript deals with the pipeline of methods for mosaicking processing for push-broom hyperspectral images, starting from geo-referencing up to the final spectral differences removing. The manuscript aims at illustrating both the general methodology, applicable to any UVA-based push broom hyperspectral images, and its realization on that particular sensor. However, there are still a few analysis and discussion that confusing, especially the motivation and advantage of the proposed method. Detailed comments are given as follow:

  • The title of the manuscript is “Mosaicking UAV-based push-broom hyperspectral images for complex environment monitoring”, it is not clear what complex environment is? I don’t think urban scape, river course and forest are complex environment.
  • What is the innovation of the proposed mosaicking methodology? Do all mosaicking methods include geometric rectification, optimal band selection, image registration and image fusion steps? What is the difficult for the mosaicking chain? How does the proposed method solve those problem?
  • In table 2, different side overlap rates are set for different study area. In line 512, it illustrates that your work focuses on the image mosaicking with narrow sideward overlaps between image strips. For this purpose, different overlap rates should be considered to verify that the proposed method has a superior effect on small overlap rate.
  • There are three methods adopted for image registration, why do you choose these three methods? Table 4, 5, 6 showed that the SIFT and RANSAC-based method gets good image registration result for forest, river and urban scape. However, there are not enough analysis for the image registration results.
  • From line 331 to line 356, there are many functions and parameters such as . All those functions should be explained. These meaning and expressions are very confusing.

Author Response

Dear Reviewer

         Thank you for your comments, I have revised the paper according to your suggestions. Please see the attached document and the revised paper for the details. I am looking forwards for your reply. 

Reviewer 2 Report

The manuscript "Mosaicking UAV-based push-broom hyperspectral images over complex environments" is submitted as a full manuscript to the Remote Sensing (MDPI) journal. The manuscript is led by Lina Yi and five co-authors from four institutions indicating a good network and research collaboration. The manuscript presents an interesting research application for those researchers using hyperspectral sensors adapted in UAV systems. The research was evaluated in different environments what brings the possibility of comparing results. I lovely read the manuscript and would suggest some changes in order to improve quality. My major concern is the absence of a proper study area location and a better detail of each selected environment. Additionally, the parameters and flight settings were also not the same over the environments what could be better explained. Interestingly, it is also important to mention that hyperspectral systems are still an emerging technology and we still do not have it popularized in science for the majority of the publications due to high cost. However, it is known that some cameras make use of Bayer filters to get near close products at certain wavelengths but they will never have the spectral sensibility of push-broom systems. This is an aspect that is still neglected and we see several researchers performing their studies without taking those intrinsic issues into account. This aspect is something that could be incorporated and would surely bring some good contribution to science. Another important aspect that was also somehow neglected in this manuscript and could be mentioned is the fact that some systems only covered certain spectral intervals and the operator needs a second system to cover a larger spectral interval. As a consequence, some mismatches may occur requiring a proper coregistration procedure that is also not that usual and known. Finally, some spectral noise such as smile effects and radiometric distortion could be also be explored and mentioned besides topographic effects.

In summary, I recognize the efforts of the authors and that the presented research already deserves publication. However, it would be worth at least mentioning those above-mentioned aspects and most importantly, in which system the reader can benefit from the proposed methods? Is the presented report and procedures expected to be available in a toolbox? I am not confident that usual researchers are able to implement the procedures that make future citation and reproducible difficult. In this sense, it would be very important to consider such aspects in the revised version of the manuscript. Below some comments that I judge also important to consider.

L19-: add "although not frequent/usual or emerging";

L33: please rewrite and explain briefly why;

L58: please add some comments about the systems available;

L76-: please compare other systems too;

L102-: i would also mention texture;

L118: please add in a proper way the reference. Usually, we add the author's name;

L145-149: please state it better;

L159: briefly explain some aspects of the legislation for the UAV flights or provide some additional references;

L195-: please add a proper study area location map;

L241: please provide some subsets and show the local features;

L257: please increase the figure and add the subsection numbers on it;

L368: please provide some additional information on where these concepts can be applied and provide some hints for those who are interested in replying your experiment;

L489: please add some subsections to the discussion and also explore better the findings with the literature;

L557: please shorten the conclusions and do not add references to them. It is better to bring the discussion in the discussion section;

In summary, it is an interesting application. Thank you for the opportunity to evaluate this research application.

Author Response

Dear reviewer:

        We appreciated your commends. We have revised the paper according to your suggestions. Please see the attached document and the revised paper. We are looking forwards to your reply. 

Round 2

Reviewer 2 Report

The manuscript "Mosaicking UAV-based push-broom hyperspectral images over complex environments" is a resubmission of previous work. The revised version of the manuscript presents an interesting research application for those researchers using hyperspectral sensors adapted in UAV systems. 

In general, I do like reading again the manuscript and I do confirm that most of the comments and concerns were properly addressed and implemented in the revised manuscript. However, since the research was evaluated in different environments I would lovely ask to add a proper study area location. Sorry for insisting on this, but I believe it shall appear before that photograph of the LIDAR system.

Additionally, there are still some minor issues with the reference style during the writing of your text. Please check the proper style required by the journal and the way of citing it NAME [NUMBER] rather than the year.

Please also check again labels, legend, and text of the axis of all figures in another and proper software. I believe that prints captured by ENVI (or similar digital image processing software) are not good. Please export the information and edit them in other software (Excel, R or Grapher, Origin, etc). In general, thank you for the opportunity to evaluate this interesting research. 

Author Response

Dear reviewer:

        Thank you very much for your suggestions. We have revised the paper accordingly. The references and figures and some of the minor issues were all revised to meet the requirements of the journal. Please see the attached "Cover letter 3" for the details. 

Best regards. 
